# Non-canonical signalling mediates changes in fungal cell wall PAMPs that drive immune evasion

Arnab Pradhan[1,2,5], Gabriela M. Avelar[1,5], Judith M. Bain[1], Delma Childers[1], Chloe Pelletier[1], Daniel E. Larcombe[1,2], Elena Shekhova[1,2], Mihai G. Netea[3], Gordon D. Brown[1,2], Lars Erwig[1,4], Neil A.R. Gow[1,2] & Alistair J.P. Brown [1,2]*

To colonise their host, pathogens must counter local environmental and immunological challenges. Here, we reveal that the fungal pathogen *Candida albicans* exploits diverse host-associated signals to promote immune evasion by masking of a major pathogen-associated molecular pattern (PAMP), β-glucan. Certain nutrients, stresses and antifungal drugs trigger β-glucan masking, whereas other inputs, such as nitrogen sources and quorum sensing molecules, exert limited effects on this PAMP. In particular, iron limitation triggers substantial changes in the cell wall that reduce β-glucan exposure. This correlates with reduced phagocytosis by macrophages and attenuated cytokine responses by peripheral blood mononuclear cells. Iron limitation-induced β-glucan masking depends on parallel signalling via the iron transceptor Ftr1 and the iron-responsive transcription factor Sef1, and the protein kinase A pathway. Our data reveal that *C. albicans* exploits a diverse range of specific host signals to trigger protective anticipatory responses against impending phagocytic attack and promote host colonisation.

[1] Institute of Medical Sciences, Foresterhill, Aberdeen, UK. [2] Medical Research Council Centre for Medical Mycology, School of Biosciences, University of Exeter, Geoffrey Pope Building, Exeter EX4 4QD, UK. [3] Department of Internal Medicine and Radboud Center for Infectious Diseases, Radboud University Medical Center, Nijmegen, Netherlands. [4] Experimental Medicine, Galvani Bioelectronics, Stevenage SG1 2NY, UK. [5] These authors contributed equally: Arnab Pradhan, Gabriela M. Avelar. *email: a.j.p.brown@exeter.ac.uk

Many microbes inhabit complex and dynamic niches that demand a high degree of environmental flexibility. Consequently, microbes have evolved specialised adaptive responses that counter environmental challenges, such as exposure to reactive chemical species, or changes in ambient pH, temperature, and osmolarity[1,2]. Furthermore, some microbes that inhabit reasonably predictable environments, where one specific challenge is often followed by a second type of challenge, have evolved elegant anticipatory responses whereby the first input triggers a pre-emptive response that protects the organism against the impending second challenge[3,4].

The immune system represents a major obstacle for most pathogenic microbes as well as colonising microbial flora. This is certainly the case for opportunistic fungal pathogens such as Candida albicans, Cryptococcus neoformans and Aspergillus fumigatus, which cause over one million life-threatening systemic infections per annum[5]. C. albicans is particularly interesting because this potentially deadly pathogen has achieved commensal status, colonising the gastrointestinal tracts of the majority of healthy individuals. Clearly, C. albicans must have evolved effective immune evasion strategies[6,7], as well as robust environmental nutrient and stress responses[8,9], which permit colonisation of an immunologically competent host. Here, we show that C. albicans has evolved anticipatory responses that link immune evasion with environmental adaptation.

Our innate immune system recognises fungal cells as foreign agents by detecting specific fungal pathogen-associated molecular patterns (PAMPs). The major fungal PAMPs, β-glucan, mannan and chitin, are critical components of the fungal cell wall and, consequently, are exposed at the fungal cell surface[10]. Myeloid cells detect these PAMPs via cognate receptors, termed pattern recognition receptors (PRRs)[10,11], and the recognition of fungal β-glucan by the receptor Dectin-1 plays a major role in anti-C. albicans immune responses. Polymorphisms that attenuate Dectin-1 functionality in humans are associated with altered cytokine responses to C. albicans and elevated susceptibility to recurrent mucocutaneous candidiasis and gut colonisation[12,13]. In mice, the inactivation of Dectin-1 decreases inflammatory responses against C. albicans and increases fungal colonisation during systemic, gastrointestinal and mucosal infections[14–16]. Interestingly, the strength of the Dectin-1 knockout phenotype can depend on upon C. albicans adaptation in vivo[15–17].

Recognition of β-glucan by Dectin-1 activates myeloid cell signalling, fungal phagocytosis and the production of proinflammatory cytokines. The macrophages and neutrophils then attempt to kill the fungus with reactive oxygen and nitrogen species (ROS and RNS) and cation fluxes[18]. The fungus normally responds to these stresses by activating robust oxidative, nitrosative and cationic stress responses[8,19,20]. However, combinations of these stresses kill fungal cells effectively[21]. Therefore, immune evasion strategies that attenuate fungal recognition and phagocytosis would present the fungus with an advantage during its interactions with innate immune cells.

Examples of fungal immune evasion include the RodA hydrophobin-mediated masking of melanin and β-glucan on the A. fumigatus spore surface[22], the synthesis of an outer polysaccharide capsule by C. neoformans to mask its cell wall β-glucan[23], and the production of α-glucan and expression of the Eng1 β-glucanase by Histoplasma capsulatum to reduce β-glucan exposure at its cell surface[24]. C. albicans exposes more β-glucan at its cell surface during hyphal development, systemic infection, and in response to the acidic pH associated with vulvovaginal niches[25,26]. On the other hand, exposure to lactate or hypoxia triggers β-glucan masking in C. albicans, which leads to decreased phagocytic recognition and altered cytokine responses[27,28].

Here we reveal that C. albicans has evolved to exploit additional host inputs to modulate β-glucan exposure at its cell surface, thereby affecting the anti-Candida cytokine responses of innate immune cells. We examined the effects of iron limitation because this condition triggers strong β-glucan masking, and because iron acquisition and homoeostasis are critical for fungal virulence[29–31]. The host imposes nutritional immunity upon the fungus, whereby immune infiltrates reduce the local availability of iron in an attempt to deprive the fungus of this essential micronutrient[31]. In turn, C. albicans responds by activating efficient iron scavenging mechanisms and moderating the expression of iron-demanding functions[30–32], some which are essential for virulence[29]. Iron acquisition and homoeostasis are tightly controlled in C. albicans via an evolutionarily conserved regulatory circuit that includes the transcriptional repressors Sfu1 and the transcriptional activator Sef1 (refs. [33,34]). We reveal additional signalling mechanisms that are essential for iron limitation-induced β-glucan masking. We also show that this phenotype promotes immune evasion in C. albicans.

## Results

**Host signals affect β-glucan exposure in C. albicans.** Using a combination of flow cytometry and microscopy, we first tested whether specific host inputs affect the levels of β-glucan exposure at the C. albicans cell surface. We selected host inputs that are known to induce significant adaptive responses in C. albicans, first testing changes in carbon or nitrogen source. Exponentially growing cells were allowed to adapt to each new condition for 5 h, fixed, and then their levels of β-glucan exposure were quantified by flow cytometry after staining with Fc-Dectin-1, relative to the control condition (Fig. 1a). The cells were also visualised by fluorescence microscopy (Fig. 1b). Cells grown on lactate displayed β-glucan masking, which recapitulated our previous observations[27]. β-glucan masking was also observed for cells grown on glycerol (Fig. 1c). In contrast, levels of β-glucan exposure increased following growth on other carbon sources, such as acetate and butyrate (Fig. 1a–c), which are abundant in the gastrointestinal tract[35]. Meanwhile, β-glucan exposure was not significantly affected by the changes in nitrogen source we examined (Fig. 1).

Before extending our screen, we checked conditions that have been examined previously. We observed significant β-glucan exposure at pH4 (Fig. 2a), which was consistent with the work of Sherrington et al.[26] who reported that acidic pH induces β-glucan exposure. We also recapitulated our previous observation[28] that hypoxia triggers β-glucan masking in C. albicans (Fig. 2b). Following these additional validations of our screen, we then examined the effect of depleting essential micronutrients. Depriving the fungus of manganese or zinc led to increased β-glucan exposure at the cell surface, whereas copper or iron limitation led to strong β-glucan masking (Fig. 2c). Changing the ambient temperature to between 25 and 37 °C, or treating the C. albicans cells with an osmotic or oxidative stress or with quorum sensing molecules, had no significant effect upon β-glucan exposure (Fig. 2d–f). However, elevating the ambient temperature to 42 °C, treating with a cell wall stress (calcofluor white (CFW)) or with sub-inhibitory concentrations of antifungal drug triggered β-glucan exposure (Fig. 2d, e). Wheeler and co-workers previously reported that Caspofungin treatment enhances β-glucan exposure[25,36]. We extended this by showing that other classes of antifungal drug also trigger significant increases in β-glucan exposure (Fig. 2e), whilst quorum sensing molecules do not (Fig. 2f). Therefore, a variety of cell wall-related stresses affect β-glucan exposure. Most significantly, we conclude that C. albicans exploits a diverse range of physiologically relevant

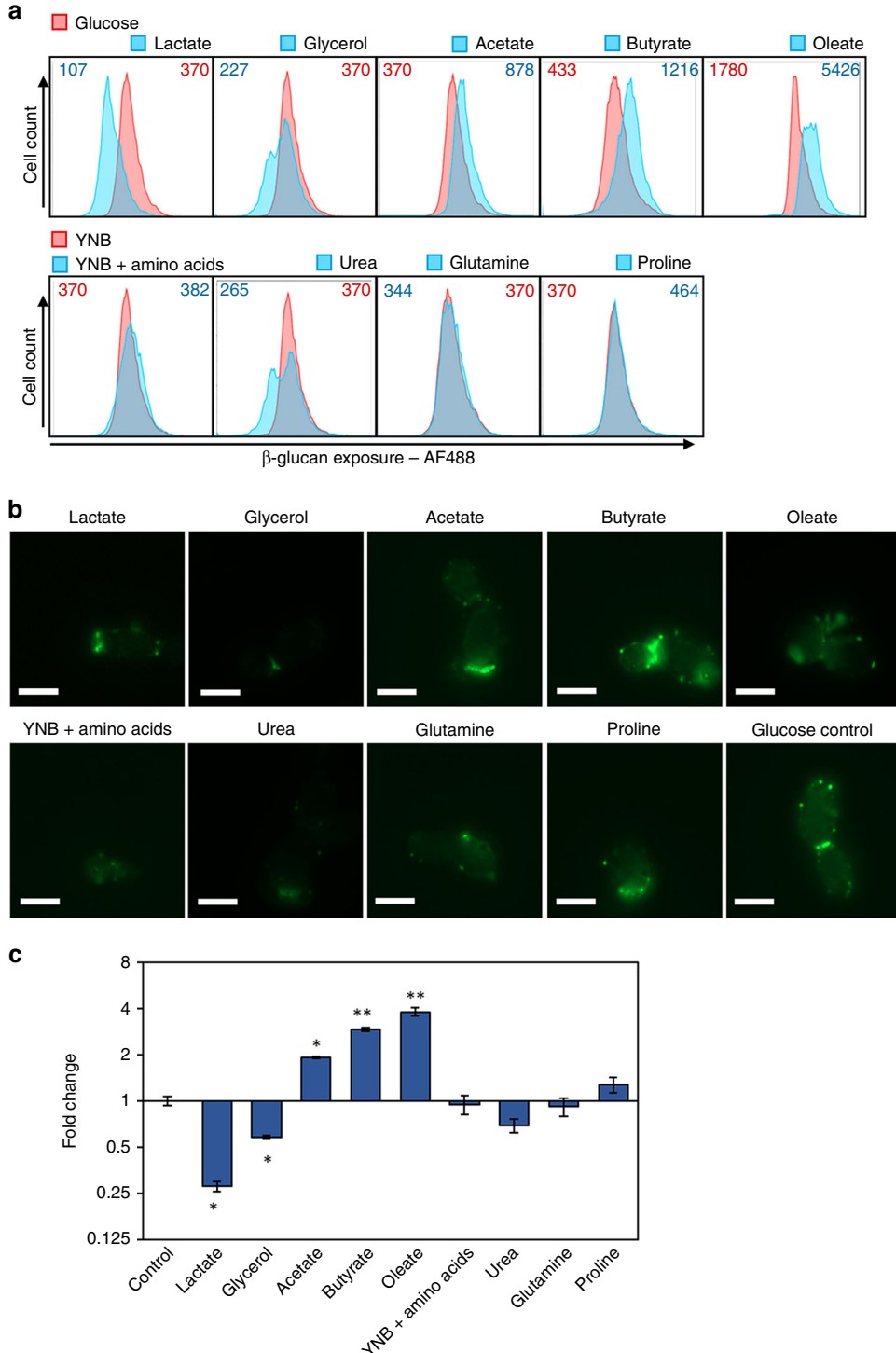

**Fig. 1** Certain nutrients impact β-glucan exposure at the *C. albicans* cell surface. *C. albicans* SC5314 cells (Supplementary Table 1) were grown overnight in minimal medium containing glucose and ammonia as sole carbon and nitrogen sources, respectively (YNB). They were then transferred to fresh YNB media containing different carbon or nitrogen sources, as shown, and grown for a further 5 h. These cells were then fixed and stained with Fc-dectin-1 to examine β-glucan exposure at their cell surface. **a** Their β-glucan exposure was quantified by flow cytometry (Supplementary Fig. 4). One of three independent experiments are shown for each condition. The median fluorescence intensity (MFI) for each population is indicated: pink, glucose and YNB controls for changes in carbon and nitrogen sources, respectively; cyan, alternative carbon or nitrogen source. **b** Fluorescence microscopy of cells from the same populations stained for exposed β-glucan (Fc-dectin-1; green): scale bars, 5 μm. **c** The fold change in β-glucan exposure for these cell populations calculated relative to the control cells grown with glucose and ammonia. Means and standard deviations from three independent replicate experiments are shown, and the data were analysed using ANOVA with Tukey's multiple comparison test: *$p < 0.05$; **$p < 0.011$. Source data are provided as a Source Data file.

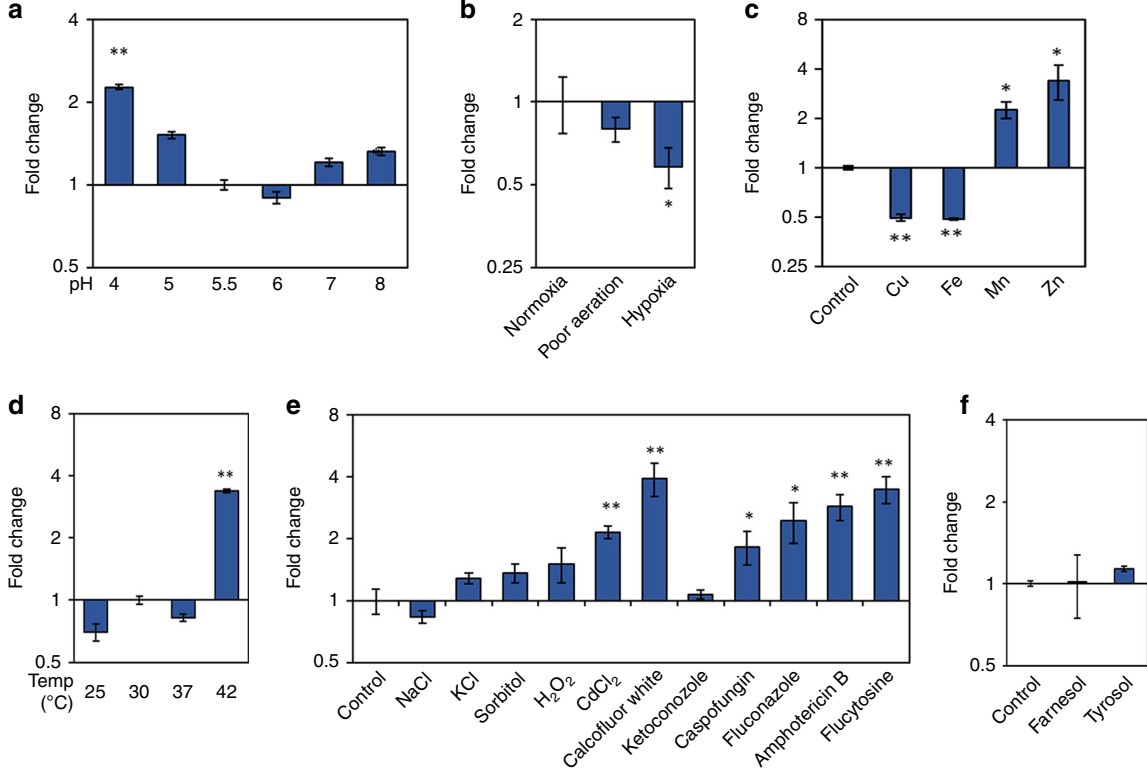

**Fig. 2** Specific environmental inputs affect β-glucan exposure in *C. albicans*. *C. albicans* SC5314 cells (Supplementary Table 1) were grown overnight in minimal medium, transferred to fresh medium containing a new environmental challenge, and grown for a further 5 h. The cells were then fixed, stained with Fc-dectin-1, and their levels of β-glucan exposure quantified by flow cytometry (Supplementary Fig. 4). The fold change in β-glucan exposure was calculated relative to the control cells in minimal medium. For most conditions the control medium was SD, but for micronutrient depletion the control medium was LXM. **a** Impact of adaptation to different ambient pHs on β-glucan exposure. **b** Effect of changing oxygen levels. **c** Impact of depletion for specific micronutrients. **d** Effect of changing the ambient temperature. **e** Impact of specific environmental stresses: 0.5 M NaCl; 0.3 M KCl; 1.2 M sorbitol; 5 mM $H_2O_2$; 0.5 mM $CdSO_4$; 200 μg mL$^{-1}$ congo red; 50 μg mL$^{-1}$ calcofluor white; 0.032 μg mL$^{-1}$ ketoconazole; 0.032 μg mL$^{-1}$ caspofungin; 0.25 μg mL$^{-1}$ fluconazole; 0.25 μg mL$^{-1}$ amphotericin B; 0.13 μg mL$^{-1}$ flucytosine. **f** Effect of quorum sensing molecules; 50 mM farnesol; 50 mM tyrosol. Means and standard deviations from three independent replicate experiments are shown, and the data were analysed using ANOVA with Tukey's multiple comparison test: *$p < 0.05$; **$p < 0.01$. Source data are provided as a Source Data file.

inputs, which include short chain fatty acids, hypoxia and iron depletion, to activate masking of a major immunostimulatory PAMP at its cell surface.

**β-glucan exposure affects host responses to *C. albicans*.** To test whether the observed changes in β-glucan exposure influence innate immune responses, we quantified the levels of TNF-α, RANTES, MIP-1α, IL-6, and IL-10 produced by human peripheral blood mononuclear cells (PBMCs) after contact with *C. albicans* cells that had adapted to each of the 39 conditions examined above. Despite the diversity of conditions examined, we observed highly significant degrees of correlation between the levels of β-glucan exposure and the amounts of TNF-α, RANTES, IL-10, and MIP-1α generated by the PBMCs (Fig. 3). The correlation with IL-6 levels was less strong, but statistically significant nevertheless. These data, which are consistent with the immunomodulatory effects of β-glucan[37–40], strongly suggest that the changes in β-glucan exposure, driven by the adaptive responses to the growth conditions tested, affect innate immune responses against *C. albicans*.

**Iron depletion activates β-glucan masking.** Our screen revealed iron depletion as triggering one of the strongest β-glucan masking responses we observed for *C. albicans* (Fig. 2c). Iron limitation is known to be a significant host input signal during systemic fungal

infection, and the activation of robust iron scavenging mechanisms is essential for fungal pathogenicity[29–31]. Therefore, we examined this iron limitation-induced β-glucan masking phenotype in more depth.

The growth of *C. albicans* under iron-limiting conditions led to significant changes in the cell wall (Fig. 4a). Quantification of transmission electron micrographs of iron-limited cells revealed that their cell walls, and particularly the inner glucan–chitin layer of these walls, were significantly thicker than those of the control iron replete cells (Fig. 4a, b). This correlated with a significant decrease in β-glucan exposure in these iron-limited cells, as revealed by fluorescence microscopy and flow cytometry (Fig. 4c–e). This strong β-glucan masking (Fig. 4e) reinforced the findings of our screen (Fig. 2c). Masking involved decreased β-glucan exposure at punctate sites that decorate the *C. albicans* cell surface, as well as on bud scars (Fig. 4c). This iron limitation-induced β-glucan masking phenotype was observed for clinical isolates from each of the four major epidemiological clades of *C. albicans* (Fig. 4f), indicating that this phenotype is displayed by diverse clinical isolates.

We examined the effects of iron concentration upon β-glucan masking. This revealed a threshold of about 10 μM Fe$^{3+}$, below which β-glucan masking was triggered (Supplementary Fig. 1A). Significantly, this suggests that β-glucan masking would occur at the quiescent concentration of labile iron in mammalian cells (2.5–5 μM)[41]. We also examined the rate at which β-glucan

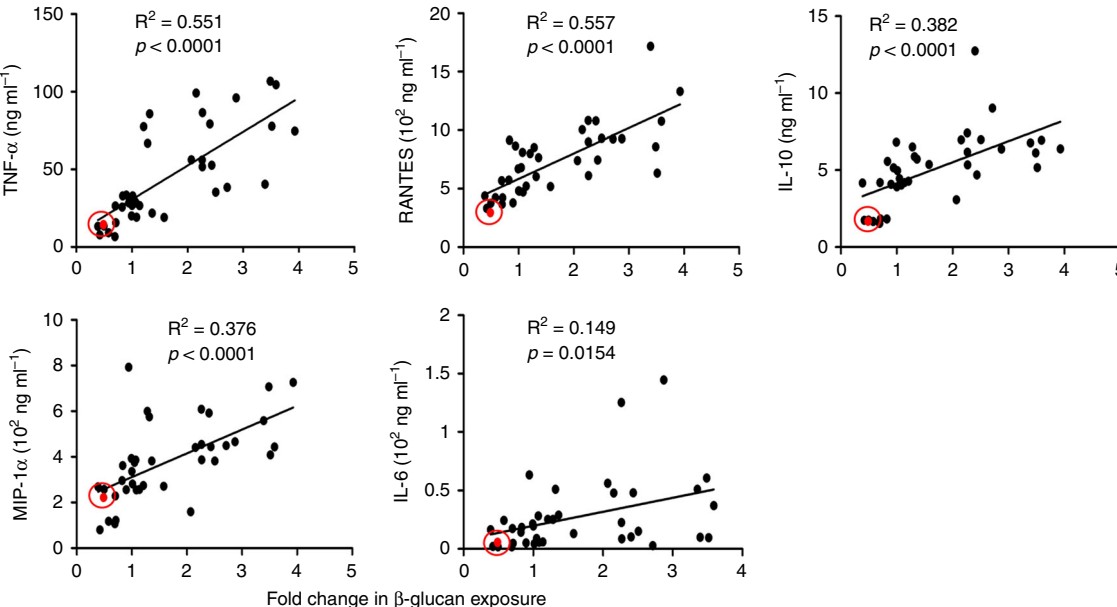

**Fig. 3** *C. albicans* β-glucan exposure correlates with PBMC cytokine responses. Wild type *C. albicans* cells (SC5314: Supplementary Table 1) were grown under the conditions described in Figs. 1, 2 and fixed. These *C. albicans* cells were then mixed with human PBMCs (duplicate samples from six different individuals) at a ratio of 5:1 (yeast:PBMCs). TNFα, RANTES, IL-10, MIP-1α, and IL-6 levels were assayed after 24 h. These data were then plotted against the fold change in β-glucan exposure triggered by each environmental condition. Each data point represents the mean of duplicate samples from four different individuals. The correlation between these parameters was tested by linear regression using the software GraphPad Prism 5. The results for iron limitation are highlighted in each graph (red dot highlighted by red circle). Source data are provided as a Source Data file.

exposure reverts after the addition of iron. β-glucan masking was lost within 2 h of iron supplementation (Supplementary Fig. 1B). This correlated with new growth of *C. albicans* (Supplementary Fig. 1), and hence the emergence of new cell wall.

**Iron limitation affects immune responses against *C. albicans*.** We tested whether the iron limitation-induced β-glucan masking phenotype influences the recognition and phagocytosis of *C. albicans* cells by innate immune cells. First, we performed time-lapse video microscopy to quantify the proportion of primary murine bone marrow-derived macrophages (BMDMs) that phagocytose masked vs. unmasked *C. albicans* cells, and also the number of fungal cells that were phagocytosed by each macrophage (Fig. 5a and Supplementary movies 1–4). The iron-limited and iron-replete *C. albicans* cells were fixed before exposure to the macrophages (see the "Methods" section), to parse out the potential confounding effects of iron limitation upon macrophage functionality or hyphal outgrowth, for example. Significantly fewer macrophages phagocytosed the β-glucan-masked *C. albicans* cells compared to their iron-replete controls. Furthermore, lower numbers of masked cells were phagocytosed per macrophage.

We then examined the chemokine and cytokine responses of PBMCs isolated from the blood of healthy human volunteers. Iron limitation-induced β-glucan masking correlated with a dramatic reduction in the amounts of TNF-α and IL-6 released by the PBMCs (Fig. 5b), and MIP-1α was also significantly reduced in response to iron-limited fungal cells. Similar trends were observed for IL-10 ($p = 0.0691$), although this change was not statistically significant. These findings were consistent with the strong effects of iron limitation upon cytokine production that we had observed during our screen of β-glucan masking conditions (Fig. 3). Our data indicate that iron limitation-induced β-glucan masking affects the responses of innate immune cells to *C. albicans*.

**β-glucan masking is dependent on PKA signalling.** Recently, we showed that hypoxia-induced β-glucan masking is dependent on cAMP-PKA signalling in *C. albicans*[28]. Therefore, we tested whether iron depletion mediates its effects on the cell wall via the same pathway. Cells that lacked either of the two catalytic subunits of PKA retained iron limitation-induced β-glucan masking, but this phenotype was absent in cells that lack both Tpk1 and Tpk2 (Fig. 6a). Interestingly, this masking phenotype was not compromised by the inactivation of adenylyl cyclase (Cyr1).

This observation contrasts with hypoxia-induced β-glucan masking, which requires both Cyr1 and PKA[28]. Previously, we suggested that lactate-induced β-glucan masking does not depend on cAMP-PKA signalling[27], but at that time PKA was viewed as essential for viability in *C. albicans*[42] and the clean *tpk1Δ tpk2Δ* double mutant was not available[43]. Therefore, we revisited the question as to whether cAMP-PKA signalling is required for lactate-induced β-glucan masking. This lactate-induced phenotype was dependent on Tpk1, not Tpk2, and was also dependent on adenylyl cyclase (Cyr1) (Fig. 6b). Therefore, lactate-induced, hypoxia-induced, and iron limitation-induced β-glucan masking all depend upon PKA, but this kinase is activated via adenylyl cyclase-independent mechanisms in response to iron limitation. This is consistent with the idea that these signals (lactate, hypoxia and iron limitation) are transduced via different upstream signalling pathways.

**Differential signalling in response to different host inputs.** Hypoxia-induced β-glucan masking is transduced via a pathway involving mitochondrial ROS[28], and intracellular iron influences ROS production via the Fenton reaction[44]. Therefore, we first examined whether intracellular ROS levels differed in iron limited and iron replete *C. albicans* cells under our experimental conditions. As expected[44], ROS levels were significantly lower in iron-depleted cells (Fig. 7a). We then tested whether any of the superoxide dismutases, Sod1–6, are required for iron

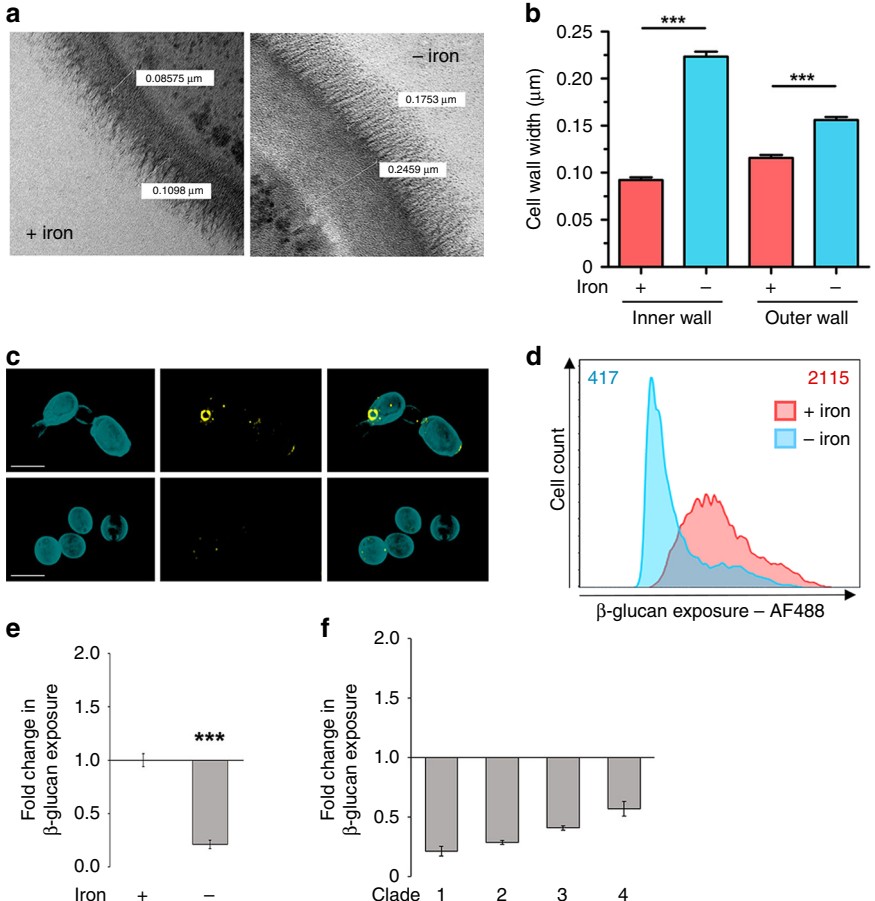

**Fig. 4 Iron limitation triggers cell wall remodelling and β-glucan masking. a** Wild type *C. albicans* cells (SC5314: Supplementary Table 1) were grown under iron replete (+ iron control) and iron-limiting conditions (− iron), and their cell walls examined by transmission electron microscopy. The thickness of the inner and outer layers of these cell walls was quantified using Image-J. **b** Means and standard deviations from images of cells (*n* = 50) from three independent replicate experiments are shown. The data were analysed using ANOVA with Tukey's multiple comparison test: ****p* < 0.001. **c** Three-dimensional images acquired using Airyscan microscopy. Yeast were grown in the presence of iron (upper panels) or the absence of iron (lower panels), and then cell wall chitin and exposed β-glucan were stained with calcofluor white (cyan) and Fc-Dectin1 (yellow), respectively: scale bars, 5 μm. **d** Analysis of β-glucan exposure on *C. albicans* SC5314 cells grown under iron replete (pink) or iron-limiting conditions (cyan) by Fc-dectin-1 staining and flow cytometry. The figure shows one representative profile for three independent experiments, and the median fluorescence intensity (MFI) for each population is indicated. **e** The fold change in β-glucan exposure for *C. albicans* SC5314 cells grown under iron-limiting conditions was calculated relative to control iron replete cells. **f** Quantification of iron-limitation-induced β-glucan masking in *C. albicans* clinical isolates: clade 1, SC5314; clade 2, IHEM16614; clade 3, J990102; clade 4, AM2005/0377 (Supplementary Table 1). Means and standard deviations from three independent replicate experiments are shown, and the data were analysed using ANOVA with Tukey's multiple comparison test: *****p* < 0.0001. Source data are provided as a Source Data file.

limitation-induced β-glucan masking. Not even Sod1, the mitochondrial superoxide dismutase essential for hypoxia-induced β-glucan masking[28], was required (Fig. 7b), suggesting that the iron-related phenotype does not depend on signalling via mitochondrial ROS. This was confirmed by our analysis of a *goa1Δ* mutant, which has lost its mitochondrial membrane potential[45] and its ability to mask in response to hypoxia[28]. The *goa1Δ* cells retained the iron limitation-induced β-glucan masking phenotype (Fig. 7b). These data show clearly that, unlike hypoxia-induced β-glucan masking, this iron-related phenotype is not dependent on signalling via mitochondrial ROS.

Lactate-induced β-glucan masking is mediated via the receptor Gpr1 and its Gα protein Gpa2 (ref. [27]). We found that iron limitation-induced β-glucan masking is not dependent on Gpr1 or Gpa2 (Fig. 8a), indicating that an alternative receptor transduces the iron signal. *FTR1* encodes a high-affinity iron permease that is essential for virulence[29]. It has been suggested that, in *S. cerevisiae*, Ftr1 is an iron transceptor that acts both as an iron transporter and an iron receptor[46], and *C. albicans FTR1*

has been shown to complement the phenotypic defects of *S. cerevisiae ftr1* cells[29]. Therefore, we tested whether iron limitation-induced β-glucan masking is dependent on Ftr1. This phenotype was blocked in *C. albicans ftr1Δ* cells (Fig. 8a), which is consistent with the idea that Ftr1 acts as an iron transceptor in this pathogen.

We then asked whether known regulators of iron homoeostasis control iron limitation-induced β-glucan masking. In *C. albicans*, iron homoeostasis is dependent on an autoregulatory circuit involving Sef1 and Sfu1 (refs. [33,34]). Sef1 is a $Cys_6Zn_2$ transcription factor that promotes iron scavenging in response to iron limitation by activating iron uptake genes. Sef1 also down-regulates *SFU1*, the role of which is to protect *C. albicans* against iron toxicity[33]. Under iron-replete conditions Sfu1 acts as a transcriptional repressor to down-regulate *SEF1* expression and it also interacts directly with Sef1 to attenuate Sef1 activity[34]. Sef1, but not Sfu1, is required for the virulence of *C. albicans*[33]. Based on their respective roles, it was conceivable that Sef1, but not Sfu1, might promote the iron depletion-related β-glucan masking phenotype.

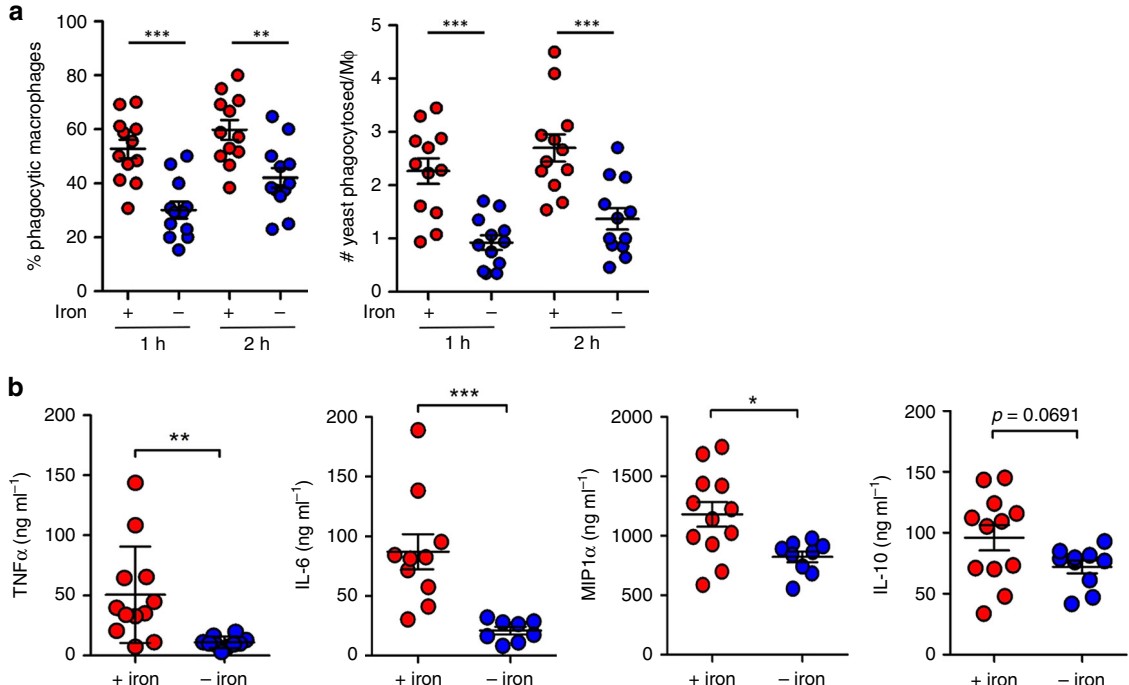

**Fig. 5** *C. albicans* adaptation to iron limitation attenuates immune responses. Wild type *C. albicans* cells (SC5314) were exposed to iron replete (red) or iron-limiting conditions (blue) for 5 h and fixed. **a** At $t = 0$, these *C. albicans* cells were mixed with murine bone marrow-derived macrophages (BMDMs) (3:1 yeast to immune cell), and the host–fungus interactions followed by time-lapse video microscopy. We quantified the proportion of BMDMs that had phagocytosed at least one *C. albicans* cell (% phagocytic macrophages) after 1 and 2 h. Also, we quantified the number of *C. albicans* cells phagocytosed by each BMDM at these time points. Each point represents the mean of data from one movie (four movies per mouse; $n = 3$ mice). **b** The cytokine responses of human PBMCs to the iron replete and iron deplete *C. albicans* cells was examined by mixing them with human PBMCs (5:1 yeast to immune cell; duplicate samples from four different individuals) and assaying TNFα, IL-6, MIP-1α, IL-10, and RANTES levels after 24 h. Means and standard deviations are shown, and these data were analysed using ANOVA with the Bonferroni post-hoc test: *$p < 0.05$; **$p < 0.01$; ***$p < 0.001$. Source data are provided as a Source Data file.

We tested this hypothesis and, as predicted, iron limitation-induced β-glucan masking was blocked in *sef1Δ* cells, but not *sfu1Δ* cells (Fig. 8b).

**Parallel pathways drive iron-related β-glucan masking**. Based on the above findings we concluded that Ftr1, Sef1 and PKA are required for iron limitation-induced β-glucan masking. We examined the relationship between Ftr1 and PKA signalling by asking whether the β-glucan masking defect of *ftr1Δ* cells could be suppressed by activating PKA via the addition of exogenous cAMP. As described previously[28], we used dibutyryl-cAMP (db-cAMP), because this cAMP derivative is membrane-permeable. The db-cAMP was able to suppress the hypoxia-induced masking defect of *goa1* cells (Supplementary Fig. 2), thereby recapitulating our previous observations[28]. However, it did not suppress the iron limitation-induced β-glucan masking defect of *ftr1Δ* cells (Fig. 9a), suggesting that Ftr1 and PKA act on parallel pathways to promote iron limitation-induced β-glucan masking.

We reasoned that this masking phenotype might be mediated by a combination of cell wall synthesis, as well as cell wall remodelling because iron depletion leads to a dramatic increase in cell wall thickness (Fig. 4a, b). The increase in chitin associated with this elevated cell wall biomass was examined by staining with CFW. As expected (Fig. 4a), wild type *C. albicans* displayed a dramatic increase in CFW staining following iron limitation, as revealed by fluorescence microscopy and flow cytometry (Fig. 9b, c). Quantification of the fluorescence intensity across cells revealed that iron depletion led to increases in both the fluorescence intensity and thickness of the CFW-stained layer in wild type cells (Fig. 9b), which was consistent with the increase

in their inner cell wall observed by TEM (Fig. 4a, b). The large errors in fluorescence measurements reflected the population heterogeneity of CFW-stained cells observed by cytometry (Fig. 9c). We took advantage of these dramatic changes in CFW staining to assess the contributions of PKA (Tpk1 and Tpk2) and Ftr1 to the changes in cell wall architecture mediated by iron limitation. Inactivating Ftr1, but not PKA, blocked this effect (Fig. 9b, c), suggesting that Ftr1 is essential for the increase in cell wall biomass in response to iron limitation. In contrast, the role of PKA appears to lie in cell wall remodelling, since this signalling pathway mediates β-glucan masking under conditions where cell wall biomass decreases[28].

We then asked whether Sef1 lies on the PKA remodelling pathway or the Ftr1 cell wall biosynthesis pathway. *C. albicans* *sef1Δ* cells displayed a similar cell wall defect to the *ftr1Δ* mutant in that no increase in CFW staining was observed following iron limitation (Fig. 9c). In contrast, the *tpk1Δ tpk2Δ* mutant behaved similarly to wild type cells. Also, *sef1Δ* and *ftr1Δ* cells showed the expected growth defects on low iron media, given the roles of Sef1 and Ftr1 in iron acquisition[29,33]. Meanwhile, the growth of *tpk1Δ tpk2Δ* cells was not further compromised on low iron media (Fig. 9d). These data are consistent with the idea that the iron transceptor Ftr1, together with the iron limitation-responsive transcription factor Sef1, lie on a parallel signalling pathway to PKA.

## Discussion

The last decades have seen dramatic advances in our understanding of antifungal immunity and, in particular, the mechanisms by which innate myeloid cells recognise invading

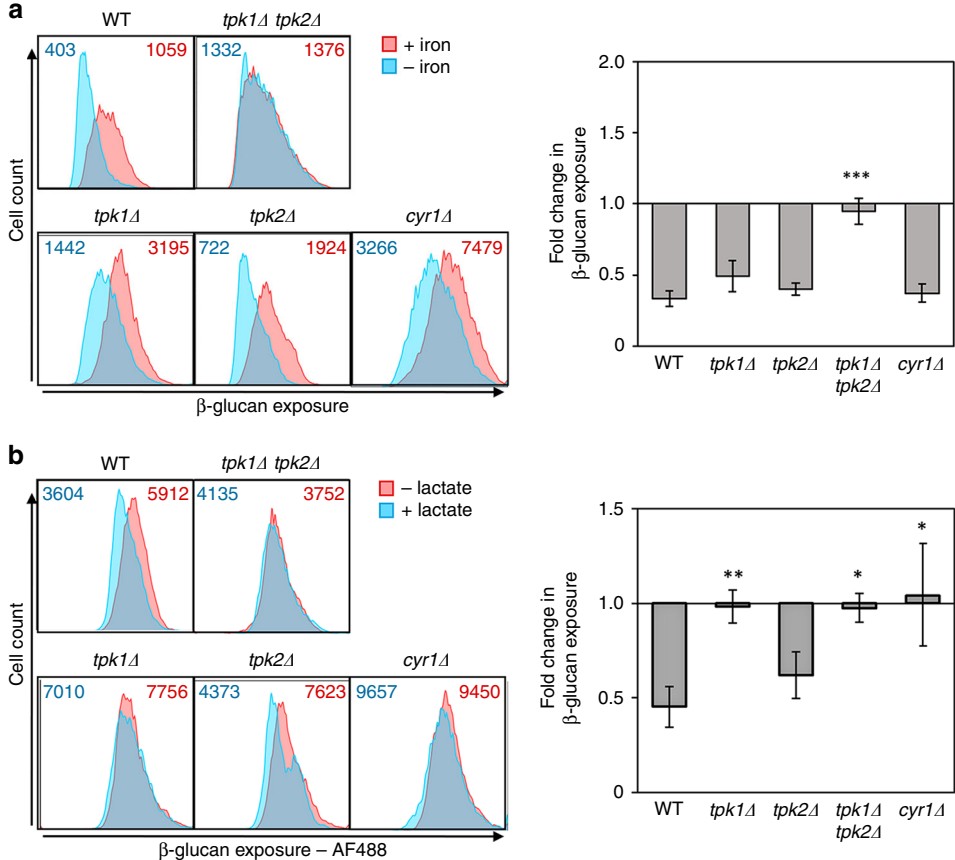

**Fig. 6** Protein kinase A signalling mediates iron-limitation-induced β-glucan masking. **a** *C. albicans* strains were exposed to iron replete (pink) or iron deplete conditions (cyan) for 5 h, fixed and stained with Fc-dectin-1, and their levels of β-glucan exposure quantified by cytometry: WT, wild type, SN152; *tpk1Δ*; *tpk2Δ*; *tpk1Δ tpk2Δ*; *cyr1Δ*, CR323 (Supplementary Table 1). Median fluorescence intensities (MFIs) for the iron replete and iron deplete cell populations are shown. The right-hand panel shows the fold change in β-glucan exposure for each strain, calculated by dividing the MFI under iron-limiting conditions by the MFI for the corresponding control iron replete cells. **b** This experiment was performed for the same *C. albicans* strains grown on SD and exposed for 5 h to 0% or 2% lactate. Means and standard deviations from three independent replicate experiments are shown, and the data were analysed using ANOVA with Tukey's multiple comparison test: *$p < 0.05$; **$p < 0.01$; ***$p < 0.001$. Source data are provided as a Source Data file.

fungal cells and thereby trigger innate and, ultimately, adaptive immune responses against these fungi[47,48]. Numerous studies highlighted in these reviews have revealed the key roles of C-type lectin receptors (CLRs) and Toll-like receptors (TLRs) in fungal recognition. Many of these receptors have been shown to associate with PAMPs at the *Candida* cell surface, such as β-glucan (e.g. Dectin-1, CD23, CD36, CR3, TLR2) or α-mannan (Dectin-2, Dectin-3, Mincle, FcγR, CD23, CR3, TLR2, TLR4, TLR6)[48]. The issue is that, for many of these experiments, the *Candida* cells were grown under standardised growth media that differ significantly from most host niches. Yet it is becoming clear that *C. albicans* alters its cell surface as it undergoes environmental adaptation. Previously it was reported that exposure to Caspofungin or host-derived lactate, and changes in ambient pH affect the levels of β-glucan exposure at the *C. albicans* cell surface[25–27]. Our data confirm and significantly extend these studies by showing that *C. albicans* regulates the exposure of this PAMP in response to a diverse range of environmental inputs that it can experience in host niches. These include alterations in carbon source (but not nitrogen source), micronutrient availability or ambient temperature, and also exposure to stresses or to different classes of antifungal drug (Figs. 1, 2). The adaptive changes to the cell wall extend beyond alterations in β-glucan exposure. This is clearly illustrated by the changes in the overall structure of the cell wall shown in Fig. 4a, for example. Nevertheless, the changes in β-glucan exposure that we observed correlate with significant

changes in cytokine and chemokine responses to, and phagocytosis rates for, the acclimated *C. albicans* cells (Figs. 3, 5). Therefore, as *C. albicans* adapts to the dynamic niches it occupies in the host, it becomes a moving target for the immune system, and changes in β-glucan exposure would appear to make a significant contribution to this.

The nature of the signals that alter β-glucan exposure are highly significant in terms of the niches that *C. albicans* occupies in the host. This is the case for those signals that have been reported previously (e.g. lactate, ambient pH and hypoxia)[26–28,49], and this view is reinforced by the diverse classes of signal identified in the present study (e.g. other short chain fatty acids, essential micronutrients, ambient temperature and environmental stresses). For example, short chain fatty acids, such as lactate, acetate and butyrate, are abundant in the colon[35], and efficient lactate acquisition is important for fungal colonisation of this niche[50]. *C. albicans* experiences hypoxia in the colon and during the development of fungal lesions[49] and displays robust adaptive responses to this condition[51,52]. Successful adaptation to ambient pH and elevated temperatures is essential for fungal colonisation during vulvovaginal and systemic infection[53,54]. Robust responses to environmental stresses and oxidative stress, in particular, are required for fungal virulence[2,8]. Similarly, iron limitation is a significant environmental challenge for *C. albicans* during systemic infection[31], and the fungus must be able to mount robust iron scavenging responses to colonise host tissues[29]. *C. albicans*

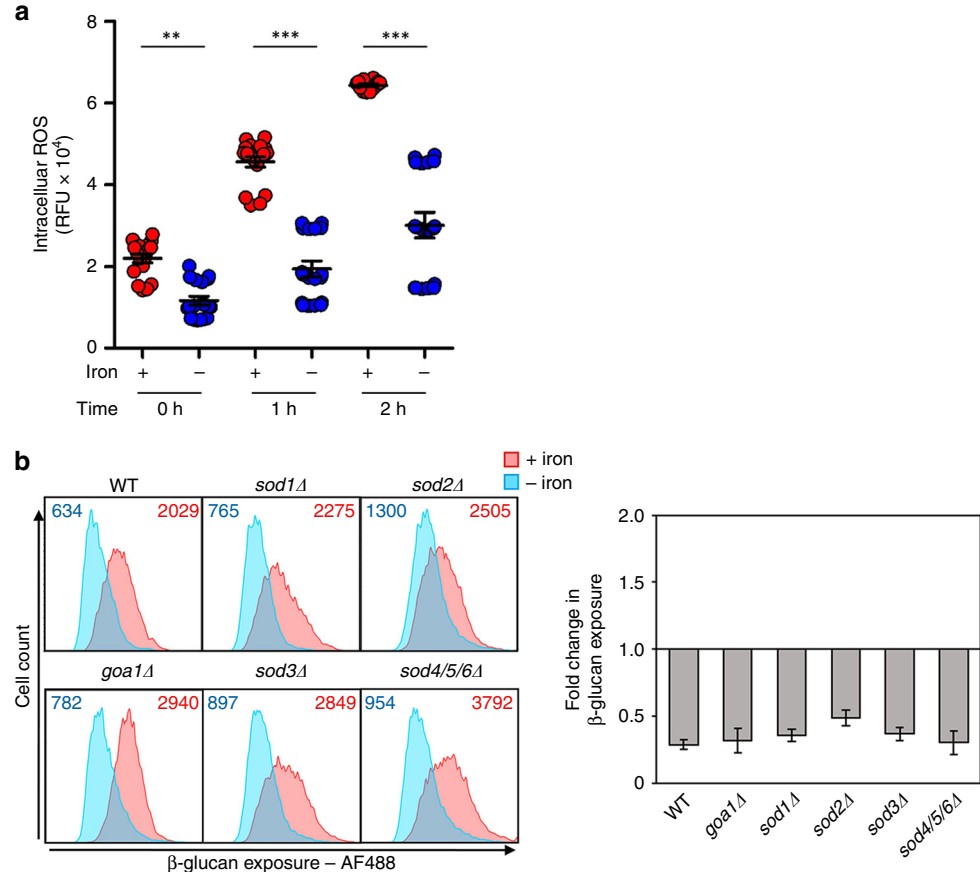

**Fig. 7** Mitochondrial ROS do not trigger iron-limitation-induced β-glucan masking. **a** The levels of intracellular ROS levels were assayed after 0, 1 and 2 h of exposure to iron replete (red) or iron-limiting conditions (blue). *C. albicans* SC5314 cells were stained with 10 μM DCFH-DA for 15 min and, after excitation at 485 nm, cellular fluorescence emission quantified at 535 nm (relative fluorescence units, RFU). The data, which represent means and standard deviations from five technical replicates from three independent experiments, were analysed using ANOVA with Tukey's multiple comparison test: *$p <$ 0.05; **$p < 0.01$; ***$p < 0.001$. **b** *C. albicans* strains were exposed to iron replete (pink) or iron-deplete conditions (cyan) for 5 h, fixed and stained with Fc-dectin-1, and their levels of β-glucan exposure quantified by cytometry: WT, SC5314; *sod1Δ*, CA-IF003; *sod2Δ*, CA-IF007; *sod3Δ*, CA-IF011 single mutants; *sod4/5/6Δ* triple mutant, CA-IF070; *goa1Δ*, GOA31 (Supplementary Table 1). Median fluorescence intensities (MFIs) for the iron replete and iron deplete cell populations are shown. The right-hand panel shows the fold change in β-glucan exposure for each strain, calculated by dividing the MFI under iron-limiting conditions by the MFI for the corresponding control iron-replete cells. Means and standard deviations from three independent replicate experiments are shown, and the data were analysed using ANOVA with Tukey's multiple comparison test. None of the changes were significant at $p <$ 0.05. Source data are provided as a Source Data file.

can scavenge iron from ferritin and transferrin via the reductive pathway involving Ftr1, from haemoglobin via the receptor Rbt5 followed by haem degradation, or by acquiring heterologous siderophores via the Sit1 transporter[30]. *C. albicans* also encounters zinc limitation during infection and activates efficient zinc scavenging mechanisms to overcome this limitation[55]. Interestingly, while iron limitation induces β-glucan masking, zinc limitation promotes β-glucan exposure (Fig. 2c). Zinc limitation also triggers the formation of large Goliath cells, whereas iron limitation does not[56]. Therefore, while nutritional immunity limits the availability of both of these essential micronutrients within host niches[31], the adaptive responses of the fungus to each are clearly different.

Clearly, the modulation of β-glucan exposure in *C. albicans* is linked to some of the most influential environmental inputs that this fungus experiences in the host. On this basis, we suggest that, as *C albicans* evolved adaptive responses to these commonly experienced environmental inputs, the fungus appears to have developed anticipatory responses that protect it against impending phagocytic attack by reducing the visibility of a key PAMP at its cell surface[57]. Naturally this phenotype may often be

considered in terms of systemic *Candida* infection. However, entry to the bloodstream of a healthy individual is generally an evolutionary dead end for most *C. albicans* cells. Therefore, regulatory links between iron depletion and PAMP masking are more likely to have evolved during interactions with the immune system in commensal niches.

The regulation of β-glucan masking in *C. albicans* appears to exploit evolutionarily conserved signalling modules (Fig. 10). Lactate-induced β-glucan masking is triggered by the receptor Gpr1, which is the closest *C. albicans* homologue to the mammalian lactate receptor[27]. Hypoxia-induced β-glucan masking is mediated by mitochondrial ROS signals, which appear to be conserved from yeasts to mammals[28]. These upstream signalling modules converge on PKA[28] (Fig. 6b), as does the iron limitation signal (Fig. 6a). The nature of the regulators that transduce this signal to PKA remain obscure, but mitochondrial ROS signalling appears to have been excluded (Fig. 7). We suggest that PKA activation mediates β-glucan masking via some form of cell wall remodelling, as this phenotype does not correlate with cell wall synthesis: hypoxia-induced β-glucan masking is associated with thin cell walls[28], whereas iron limitation-induced β-glucan

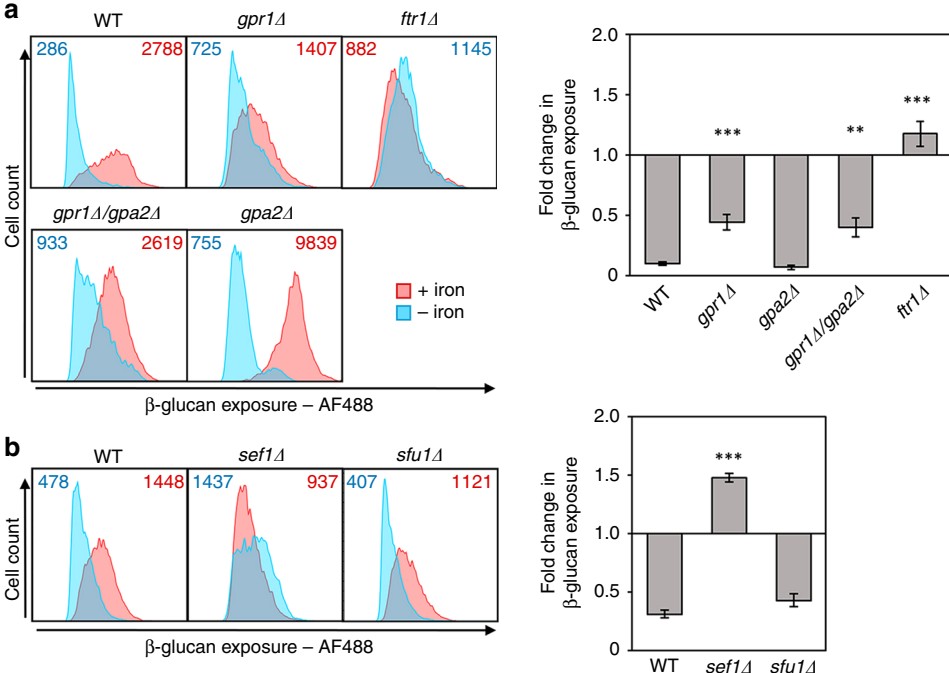

**Fig. 8** Iron-limitation-induced β-glucan masking depends on Ftr1 and Sef1. *C. albicans* strains were exposed to iron replete (pink) or iron-deplete conditions (cyan), fixed and stained with Fc-dectin-1. Their levels of β-glucan exposure were quantified by cytometry, and the median fluorescence intensities (MFIs) for the iron replete and deplete cell populations are shown. The fold changes in β-glucan exposure are shown in the right-hand panels. **a** Contributions of receptors to iron-limitation-induced β-glucan masking: WT, wild type, SC5314; gpr1Δ, LR2; gpa2Δ, NM6; gpr1Δ gpa2Δ, NM23; ftr1Δ (Supplementary Table 1). **b** Contributions of iron regulators: WT, wild type, SN152; sef1Δ; sfu1Δ (Supplementary Table 1). Means and standard deviations from three independent replicate experiments are shown, and the data were analysed using ANOVA with Tukey's multiple comparison test: *$p < 0.05$; **$p < 0.01$; ***$p < 0.001$. Source data are provided as a Source Data file.

masking is linked with much thicker cell walls (Fig. 4a, b), and lactate-induced β-glucan masking is related to cell walls with a disordered mannan outer layer[27,58].

The Ftr1 transceptor and Sef1 transcription factor are both required for iron limitation-induced β-glucan masking (Fig. 8), and both are evolutionarily conserved from yeasts to filamentous fungi[29,33] (*Candida* Genome Database, www.candidagenome. org). In contrast to the PKA module, Ftr1-Sef1 signalling does appear to be associated with new cell wall synthesis, because these iron-responsive regulators are required for the increased cell wall biomass observed after adaptation to iron depletion (Fig. 9). Increases in cell wall biomass are not the only possible means of reducing β-glucan exposure, however, as a double tpk1Δ tpk2Δ mutation blocks β-glucan masking (Fig. 6a) without inhibiting cell wall expansion (Fig. 9b, c). In addition, PKA signalling is not epistatic to Ftr1-Sef1 signalling (Fig. 9a). Therefore, we suggest that these modules lie on parallel pathways that promote β-glucan masking via different mechanisms (Fig. 10).

The mechanisms that underlie β-glucan masking in *C. albicans* appear to be complex and remain obscure. The mannan outer layer of the *C. albicans* cell wall plays an important role in masking β-glucan PAMPs from Dectin-1-mediated recognition[59]. Therefore, the dense mannan outer layer of iron-limited cells (Fig. 4a, b) is likely to contribute to their lower β-glucan exposure. However, attempts to dissect the role(s) of mannan during β-glucan masking via genetic perturbation of mannan biosynthesis have not been particularly informative[27]. For example, *MNT4* is the only predicted mannosylation gene to be induced in response to iron limitation[60], and yet *MNT4* does not appear to affect β-glucan exposure levels[61]. Iron limitation also affects the expression of a small number of cell wall carbohydrate-active enzymes[60]. Yet genes encoding β-glucan transferase (*BGL2*) and

a glycosidase (*PHR2*) are down-regulated during iron limitation, just as β-glucan masking is up-regulated. Meanwhile mutations that affect other processes, such as MAP kinase signalling, phospholipid biosynthesis or even mRNA deadenylation, can affect β-glucan exposure at the *C. albicans* cell surface[28,62,63].

Clearly β-glucan recognition is critical for antifungal immunity. Mutations that affect Dectin-1 functionality increase an individual's susceptibility to infection[12], β-glucan exposure correlates with a reduction in fitness in host niches[16], and reduced β-glucan detection promotes fungal colonisation in host niches[13,49]. Therefore, given the significance of iron limitation during fungal infection[31], we suggest that iron limitation-induced β-glucan masking influences fungal recognition during colonisation and infection. Clearly host–fungus interactions are complex and multifactorial. For example, iron limitation also affects the expression of genes encoding an adhesin (Als2), secreted aspartyl proteinases (Sap10, Sap99) and oxidative stress functions (Cat1, Grx1, Gsh2, Sod5, Tsa1)[60]. Nevertheless, β-glucan masking has been shown to promote *C. albicans* colonisation in vivo[49], and certain antifungal treatments that enhance β-glucan exposure at the *C. albicans* cell surface have been shown to influence protective immune responses against this fungus[36]. Therefore, in the future, potential inhibitors of β-glucan masking, that increase the exposure of this major PAMP at the fungal cell surface, might provide a useful means of augmenting antifungal immunotherapies.

## Methods

**Ethics statement.** We have complied with all relevant ethical regulations for work with human participants/samples. Blood was donated by healthy volunteers with their informed consent. Donors' blood was collected according to local guidelines and regulations that had been approved by the College Ethics Review Board of the

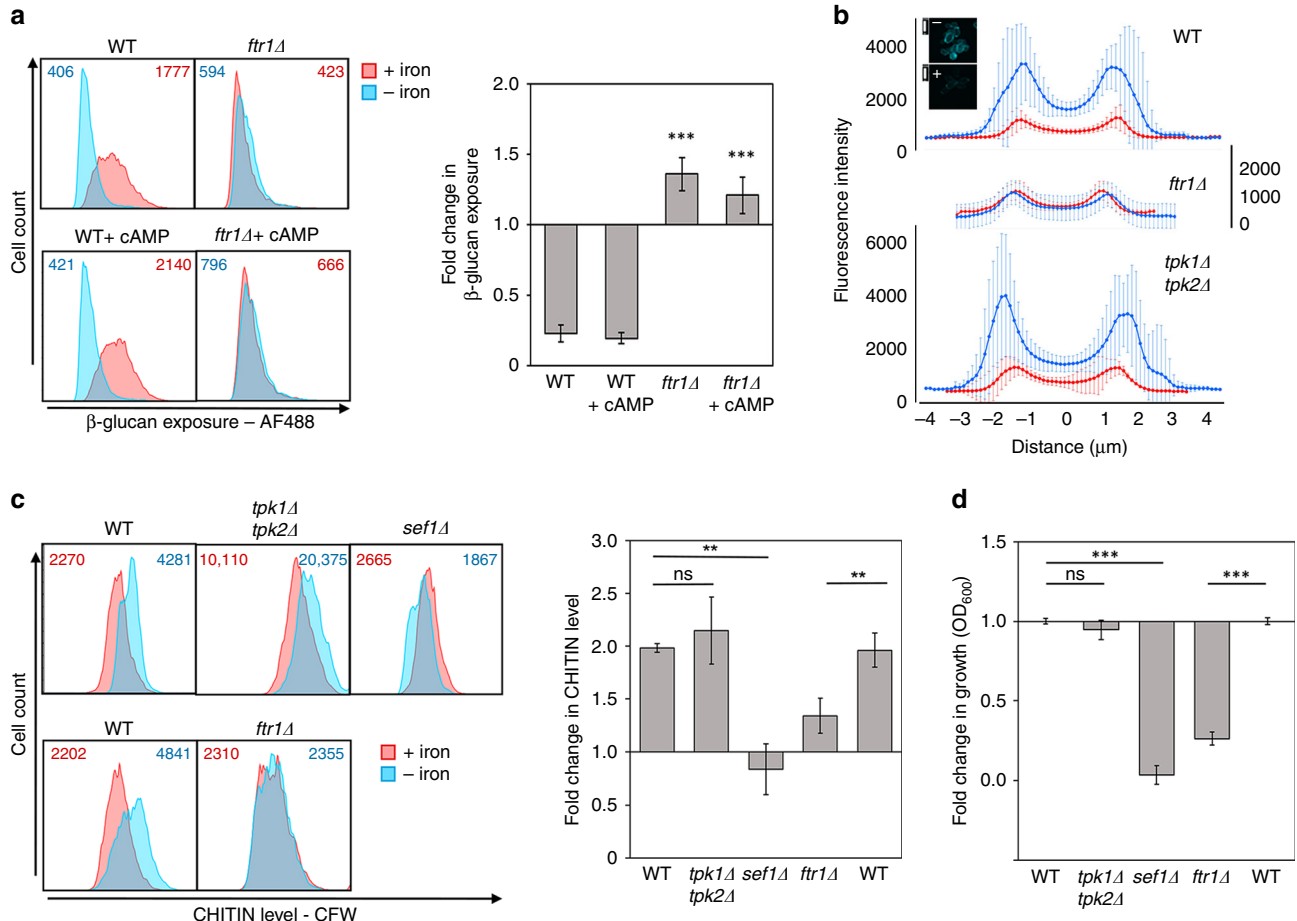

**Fig. 9** Ftr1-Sef1 and PKA induce β-glucan masking via parallel pathways. **a** Dibutyryl cAMP (cAMP) does not suppress the iron-limitation-induced β-glucan masking defect of *ftr1Δ* cells. *C. albicans* was exposed to iron replete (pink) or iron-deplete conditions (cyan) with 0 or 5 mM cAMP, and β-glucan exposure quantified (see Fig. 6): WT, wild type, SC5314; *ftr1Δ* (Supplementary Table 1). **b** The iron limitation dependent increase in cell wall thickness was examined by calcofluor white staining. *C. albicans* was grown under iron replete (red) or deplete (blue) conditions, fixed, stained with 10 μg mL$^{-1}$ calcofluor white, subjected to fluorescence microscopy (inset shows wild type +/− iron; scale bars, 5 μm: Supplementary Fig. 3). Fluorescence was quantified across individual cells (see the "Methods" section): means and standard deviations are shown for $n = 30$ cells: WT, wild type, SC5314; *tpk1Δ tpk2Δ*; *ftr1Δ*. **c** Chitin levels in these *C. albicans* cells were quantified by flow cytometric analysis of calcofluor white staining intensities. Median fluorescence intensities (MFIs) for the iron replete and deplete populations are shown. The appropriate wild type strain is shown on the left for each set of plots: top row: WT, wild type, SN152; *tpk1Δ tpk2Δ*; *sef1Δ*: bottom row: WT, wild type, SC5314; *ftr1Δ* (Supplementary Table 1). Fold changes in chitin levels are shown in the panel below the cytometry plots. **d** These *C. albicans* strains were grown overnight in LIM +/− Fe, inoculated into fresh LIM +/− Fe medium and grown for a further 5 h whereupon the $OD_{600}$ of each culture was measured. To control for strain background, the fold change in growth was determined by first calculating the effect of iron depletion for each strain ($\Delta OD_{600}^{-Fe}/\Delta OD_{600}^{+Fe}$), and then relating this to the effect in the corresponding wild type control (relative $\Delta OD_{600}^{mutant}$/relative $\Delta OD_{600}^{WT}$). For cytometry, means and standard deviations from three independent replicate experiments are shown, and the data were analysed using ANOVA with Tukey's multiple comparison test: *$p < 0.05$; **$p < 0.01$; ***$p < 0.001$. Source data are provided as a Source Data file.

University of Aberdeen (CERB/2012/11/676). All animal usage was approved by University of Aberdeen Animal Welfare and Ethical Review Body. For the preparation of BMDMs, three 7-week old male C57BL/6 mice that had been bred in-house and housed in stock cages under specific pathogen-free conditions, were selected at random. These animals did not undergo any surgical procedures prior to culling by cervical dislocation.

**Candida strains and growth conditions.** The strains used in this study are listed in Supplementary Table 1. For most experiments, *C. albicans* strains were grown overnight at 30 °C, 200 rpm in minimal medium (GYNB: 2% glucose, 0.65% yeast nitrogen base without amino acids, containing the appropriate supplements). On the day of an experiment, overnight cultures were diluted into fresh minimal medium to an $OD_{600}$ of 0.2, and incubated at 30 °C at 200 rpm for 5 h for analysis. Environmental stresses and quorum sensing molecules were added for this 5 h period at the following concentrations: 0.5 M NaCl; 0.3 M KCl; 1.2 M sorbitol; 5 mM $H_2O_2$; 0.5 mM $CdSO_4$; 200 μg mL$^{-1}$ congo red; 50 μg mL$^{-1}$ CFW; 0.032 μg mL$^{-1}$ ketoconazole; 0.032 μg mL$^{-1}$ caspofungin; 0.25 μg mL$^{-1}$ fluconazole; 0.25 μg mL$^{-1}$ amphotericin B; 0.13 μg mL$^{-1}$ flucytosine; 50 mM farnesol; and

50 mM tyrosol. Ambient pH and temperature were varied as specified. Alternative carbon sources were added at a concentration of 1% together with 1% glucose, and alternative nitrogen sources at 0.67%. To test hypoxia, cells were grown in screw cap conical flasks under nitrogen for the 5 h period, whilst the normoxic control cells were grown with aeration[28]. To test the effects of depleting specific micro-nutrients, *C. albicans* cells were grown in adapted limited X media (LXM: 11 mM glucose, 1 mM EDTA, 25 μM $FeCl_3$, 13 μM $MnSO_4$, 0.3 μM $CuSO_4$, 25 μM $ZnSO_4$, 5 mM $MgSO_4$, 1 mM NaCl, 1 mM $KH_2PO_4$, 1 mM $CaCl_2$, 10 μM $H_3BO_3$, 0.5 μM KI, 1 μM $Na_2MoO_4$, 0.4 mM uridine, 0.5 mM histidine, 0.76 mM leucine, 0.7 mM lysine, 16 nM biotin, 1.6 μM calcium pantothenate, 10 μM myo-inositol, 2 μM pyridoxine, 1 μM thiamine, 20 mM sodium citrate, 40 mM Tris–HCl pH 8) from which either copper, iron, manganese or zinc were omitted[55].

For in-depth analyses of iron limitation, the growth conditions were optimised to ensure that: (a) growth was not compromised by starving cells of this essential micronutrient during the 5 h adaptation period; and (b) pre-accumulation of large intracellular iron stores did not compromise the iron limitation over this period. These cultures were grown overnight in SD, washed in 0.5 M EDTA pH 8.0 and twice in double-distilled water to remove surface iron. Cultures were then diluted to an $OD_{600}$ of 0.2 in 5 mL limited iron medium (LIM: LXM containing 0.5 μM

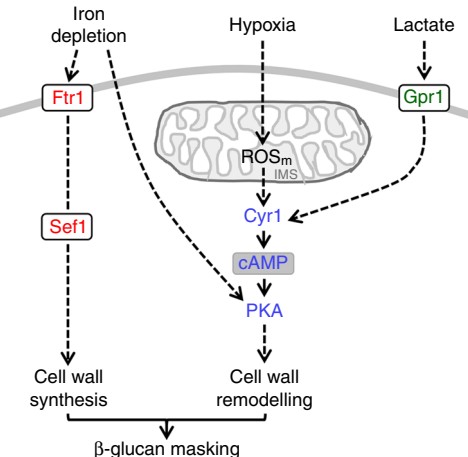

**Fig. 10** Different input-specific signalling pathways activate β-glucan masking. Protein kinase A (PKA) signalling is required for iron limitation-induced, hypoxia and lactate-induced β-glucan masking[28] (Fig. 6). However, masking in response to iron limitation is not dependent upon adenylyl cyclase (Fig. 6a), unlike lactate-induced (Fig. 6b) or hypoxia-induced β-glucan masking[28]. Regarding the upstream signalling modules, iron limitation activates β-glucan masking via the iron transceptor Ftr1 and the iron responsive transcription factor Sfe1. In contrast, lactate triggers β-glucan masking via the receptor, Gpr1 (ref. 27) and the hypoxia signal appears to be transduced via mitochondrial ROS signalling[28]. Hypoxia-induced mitochondrial ROS signalling seems to lie upstream of PKA signalling[28]. In contrast, iron-related Ftr1-Sef1 signalling appears to act independently of PKA signalling (Fig. 9). Since PKA signalling is not required for cell wall elaboration or growth under iron-limiting conditions (Fig. 9b–d) this pathway might promote β-glucan masking via cell wall remodelling mechanisms that remain to be defined. Ftr1-Sef1 signalling is required for cell wall elaboration and growth under iron limitation (Fig. 9c, d) and therefore might promote β-glucan masking by driving the synthesis of a relatively thick outer mannan layer (Figs. 4, 9). Source data are provided as a Source Data file.

FeCl$_3$) and incubated at 30 °C at 200 rpm for 24 h. Cultures were then re-diluted to an OD$_{600}$ of 0.2 into fresh LIM medium, and incubated at 30 °C at 200 rpm for 5 h to generate iron-limited cells for analysis. Control iron replete cells were for 5 h in LIM supplemented with 50 μM FeCl$_3$.

**Microscopy.** *C. albicans* cell walls were examined by high-pressure freeze substitution transmission electron microscopy. To achieve this, ultrathin sections of 100 nm thick were cut from *C. albicans* cells[58]. The sections were imaged using a Philips CM10 transmission microscope (FEI, United Kingdom) equipped with a Gatan Bioscan 792 camera. Images were recorded using a Digital Micrograph (Gatan, Abingdon Oxon, United Kingdom), and the thicknesses of the inner and outer layers of the cell wall were measured using Image-J v.1.47. An average of about 30 measurements were taken for each cell and over 30 cells were examined.

Cell surface β-glucan exposure was examined by fluorescence microscopy. Cells were fixed in 50 mM thimerosal (Sigma-Aldrich) and stained for β-glucan (1.5 μg mL$^{-1}$ Fc-Dectin-1 plus anti-human IgG conjugated to Alexafluor 488 (AF488); green). Phase differential interference contrast (DIC) and fluorescence microscopy was performed using a Zeiss Axioplan 2 microscope, the images recorded with a Hamamatsu C4742–95 digital camera (Hamamatsu Photonics, Hamamatsu, Japan), and processed using Openlab (Openlab v 4.04: Improvision, Coventry, UK).

To assay chitin levels in *C. albicans* cell walls, fixed yeast were stained with CFW (10 μg mL$^{-1}$ in phosphate buffered saline) for 30 min and washed thoroughly. Cells were mounted on glass slides in ProLong™ Diamond antifade mountant (Invitrogen) and set overnight. CFW fluorescence was imaged in 3D using a Nikon Eclipse Ti UltraVIEW VoX spinning disk microscope and Volocity software v.6.3.1 (PerkinElmer). For each condition, a fluorescent profile was determined using Volocity (line tool). Lines were drawn in *XY* at the clearest midpoint (in *Z*) for individual cells. Data were aligned using the position of double peaks, and means ± standard deviation for *n* = 30 cells were calculated.

For Airyscan imaging, yeast were stained with CFW and Dectin-1 (above), fixed and mounted to glass slides in ProLong™ Diamond antifade mountant (Invitrogen) were set overnight. A Zeiss LSM 880 Airyscan microscope with ×63 objective lens was used to acquire 3D images.

**Flow cytometry.** β-glucan exposure on *C. albicans* cells was quantified by flow cytometry[27,28]. Briefly, exponential cells were incubated for 5 h under the conditions specified and then fixed with 50 mM thimerosal (Sigma-Aldrich, Dorset, UK). The cells were then stained with Fc-Dectin-1 and anti-human IgG conjugated to Alexafluor 488 (Invitrogen), and the fluorescence of 10,000 events acquired using a BD Fortessa flow cytometer. The gating strategy and axis scales, which remained unchanged throughout, are presented in Supplementary Fig. 4. Median fluorescence intensities (MFI) were determined using FlowJo v.10 software. Each cytometry plot is representative of at least three independent biological replicates.

To examine chitin levels in *C. albicans*, thimerosal-fixed cells were stained for 60 min in the dark with 10 μg mL$^{-1}$ CFW (Sigma-Aldrich, Dorset, UK). These cells were imaged by fluorescence microscopy (above) and their fluorescence was quantified by flow cytometry without further staining (above: Supplementary Fig. 4).

**Cytokine assays.** Published procedures were used to perform the cytokine assays[27,28]. PBMCs were prepared from non-heparinised whole blood (20 mL) collected from healthy volunteers by Ficoll-Paque centrifugation according to the manufacturer's instructions (Sigma-Aldrich). Thimerosal-fixed *C. albicans* cells were washed 4× with sterile phosphate buffered saline and then incubated for 24 h with PBMCs (5:1, yeast:PBMCs). The supernatant was then collected, and specific chemokines and cytokines quantified with a Luminex® Screening kit (R&D Systems, Abingdon, UK) in the BioPlex 200 System (Bio-Rad, Watford, UK) according the manufacturer's instructions. Each data point represents the mean of duplicate samples from four different individuals.

**Live imaging of phagocytosis.** To prepare BMDMs, bone marrow was extracted from the femurs and tibias of 12-week-old male C57BL/6 mice. The BMDMs were then differentiated for 7 days[64]. *C. albicans* cells were preadapted to iron replete or limiting conditions, fixed with thimerosal and washed (as described above). The fixed fungal cells were then mixed with the BMDMs (3:1 yeast:macrophage) and imaged with a Nikon Eclipse Ti UltraVIEW VoX spinning disk microscope at 1 min intervals for up to 4 h (refs. 65,66). Using this approach, only the fungal cells, not the macrophages, were exposed to iron limitation. Movies were viewed using Volocity software and the proportion of BMDMs that were phagocytosing yeast, and the number of yeast that were phagocytosed per macrophage were quantified at hourly intervals. At each time point the differences between iron limited and replete conditions was examined using ANOVA with the Bonferroni post hoc test.

**Reactive oxygen species.** Levels of endogenous ROS in iron replete and iron deplete *C. albicans* cells (above) were assayed[67]. Briefly, fungal cells were stained with 10 μM 2′,7′-dichlorofluorescein diacetate (DCFH-DA) for 15 min at 30 °C in the dark and washing thrice in phosphate buffered saline. Following excitation at 485 nm, emission at 535 nm was measured using a TECAN SPARK fluorescence plate reader. Three independent *Candida* cultures were used, and fluorescence in each replicate was measured for five technical replicates.

**Statistical analyses.** GraphPad Prism 5 was used for statistical analyses. Data were generated from at least three independent biological replicates and then expressed as means ± standard deviation. To test the statistical difference between two sets of data with a non-parametric distribution we used one-way ANOVA (Tukey's multiple comparison test). The following *p*-values were considered: *$p < 0.05$; **$p < 0.01$; ***$p < 0.001$; **** $p < 0.0001$.

**Reporting summary.** Further information on research design is available in the Nature Research Reporting Summary linked to this article.

## Data availability

The authors declare that the data supporting the findings of this study are available within the paper (and the accompanying supplementary information files. The source data underlying Figs. 1c, 2a–f, 3, 4b, f, 5a, b, 6a, b, 7a, b, 8a, b and 9a–d, and Supplementary Figs. 1a, b and 2 are provided as a Source Data file.

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

## Acknowledgements

We are grateful to Raif Yuecel, Linda Duncan, Kimberley Sim and Ailsa Laird in the Iain Fraser Cytometry Centre, and to Kevin MacKenzie, Debbie Wilkinson, Gillian Milne and Lucy Wight in our Microscopy and Histology Core Facility for their superb support. We thank Katja Schafer and Angela Lopez for help with the design of primers and for providing CRISPR-Cas9 protocols for mutant construction. We also thank our colleagues in the *Candida* community, and in particular Jan Quinn, Guanghua Huang, Suzanne Noble, Karl Kuchler, Patrick van Dijck, Rich Calderone and Malcolm Whiteway for providing strains used in this study. This work was funded by a programme grant from the UK Medical Research Council [www.mrc.ac.uk: MR/M026663/1], and by Ph.D. studentships from the University of Aberdeen to A.P., D.E.L. The work was also supported by the Medical Research Council Centre for Medical Mycology and the University of Aberdeen [MR/N006364/1], by the European Commission [FunHoMic: H2020-MSCA-ITN-2018–812969], and by the Wellcome Trust via Investigator, Collaborative, Equipment, Strategic and Biomedical Resource awards [www.wellcome.ac.uk: 075470, 086827, 093378, 097377, 099197, 101873, 102705, 200208]. The funders had no role in study design, data collection and analysis, decision to publish, or preparation of the manuscript.

## Author contributions

A.J.P.B., N.A.R.G., L.E. and M.G.N. conceived the project. G.M.A. performed the β-glucan masking screen and cytokine assays, whilst A.P. performed most of the iron masking experiments. J.M.B., D.C., C.P., D.E.L. and E.S. provided essential experimental contributions and support such as the design of the iron-limitation growth conditions, delivery of microscopy and cytometry, iROS assays and help with data interpretation. G.D.B. provided essential materials for the assays of β-glucan exposure (Fc-Dectin-1) as well as key input to the design of the fungal immunology experiments. A.J.P.B., A.P., G.M.A., wrote the manuscript. All authors contributed to the preparation of the manuscript.

## Competing interests

The authors declare no competing interests.
