## [Peer Review File · Nature Communications]

Reviewers' comments:

Reviewer #1 (Remarks to the Author):

The manuscript by Pradhan et al describes a screen for nutrient and environmental conditions that lead to increased or decreased masking of $\beta(1,3)$ -glucan in *Candida albicans*. They make the novel observation that iron limitation, a condition that often occurs in the host environment during nutritional immune responses, leads to greater masking. They further find that this is dependent on the PKA kinases Tpk1 and Tpk2 as well as the Sef1 transcription factor acting downstream of the Ftr1 iron transporter/receptor. Finally, they show they these are acting independently of one another. This is an interesting and exciting study. There are a few points that should be addressed before it is ready for publication.

1. They refer to unmasking, or masking but in their images the changes appear to be primarily at the poles which suggests that this is occurring at the bud scars. Is it the case that the changes in unmasking are primarily revolving around the bud scars? They should use calcofluor white staining to address this and determine if their changes are primarily seen associated with the bud scars.
2. In Figure 9A they conclude the pathways are acting in parallel in part because they cannot suppress the impact of *ftr1* $\Delta\Delta$ by adding di-cyclic-cAMP to the media. However, there is not a positive control for this experiment which makes it hard to know if the cAMP is having an effect.
3. To further support the parallel argument, they should test to see if they can suppress the masking impact of *tpk1* $\Delta\Delta$ *tpk2* $\Delta\Delta$ by overexpressing Sef1.
4. In Figure 6 they measure the cell wall thickness. In Figure 9 they measure chitin production by calcofluor white staining and then imply that this indicates cell wall increases, but they need to measure this by TEM as they did in Figure 6 to be sure of this.
5. Iron depletion causes masking, but zinc depletion causes unmasking in their hands, yet both are part of nutritional immunity. They should address this paradox in their discussion.

Reviewer #2 (Remarks to the Author):

The report by Pradhan et al. is composed of two parts. First, a survey of the effects that various nutrients, chemicals, stresses and other growth conditions have on *C. albicans* β -glucan exposure. And, second, a more detailed analysis of the effect of iron depletion on β -glucan masking. In general, the data are well presented and the logic of the experiments straightforward to follow.

This manuscript is an extension of similar work on β -glucan exposure that the same group has published in the last couple of years (*Nat Microbiol* 2, 16238 (2016) and *mBio* 9, e01318-18 (2018)). The main concept, i.e. that *C. albicans* masks/exposes its β -glucan layer in response to environmental cues, has been put forward by the authors in the aforementioned publications. What is new here is the demonstration that iron depletion is another one of these signals.

Through phenotypic analyses of several deletion mutant strains, the authors convincingly demonstrate that the signaling pathway(s) involved in connecting iron availability to alterations in β -glucan exposure are not the same as the ones previously implicated in lactate- or hypoxia-mediated β -glucan modifications. However, in my opinion, it remains unclear how exactly iron depletion leads to β -glucan masking (i.e. the molecular mechanism). The authors do probe the involvement of well-known iron regulators but their analysis falls short of providing major mechanistic insights.

In terms of presentation of the story, I would recommend that the authors consider three points:

First, I think the authors overemphasize the "relevance" of the nutrients and other in vitro conditions that they evaluate to the environment inside the host. The fact of the matter is that we

do not know for sure what signals or cues are really triggering what responses in *C. albicans* when it is within host tissues (multiple signals likely act simultaneously producing complex responses). The paper reads a lot like if it was a given that lactate or iron are responsible for *Candida*'s defense against immune cells. They may be part of the answer but it is far from definitive.

Second, I find the use the word 'anticipatory' (lines 32, 53, 64, 319) troubling. I am aware that similar ideas have been posited to explain the behavior of some bacterial pathogens, but still one needs much imagination to conceive that a unicellular organism is actually able to "anticipate" something and get ready for a particular event. It is simply not a precise expression.

Third, the changes in β -glucan exposure—which the authors emphasize—appear to be only one of multiple cell wall alterations induced by iron limitation. In fact, the data presented (Fig. 4) demonstrates that iron availability alters significantly the overall structure of the *C. albicans* cell surface. Hence, contrary to what the manuscript implies, it is not completely clear that the reduction in phagocytosis of yeast cells grown under iron limitation (Fig. 5) is due exclusively to β -glucan masking. What the authors present is a correlation, which is fine. But this caveat should be acknowledged more openly and widely in the manuscript.

Minor points:

Lines 87-89: Format of references is incorrect.

Line 251: Typo: The authors probably mean 'previously' instead of 'preciously'

Reviewer #3 (Remarks to the Author):

The manuscript by Pradhan et al "Non-canonical signalling mediates changes in fungal cell wall PAMPs that drive immune evasion" describes the masking and unmasking of the major *Candida* fungal PAMP in response to different metabolic signalling pathways, with concomitant impact on host recognition of this important pathogen. This is a well-conceived and conducted study with some key findings of importance to our understanding not just of host-*Candida* interactions, but of general host-microbe interactions. This study concentrates on the pathogen response in this process, but is still of major interest to microbiologists and immunologists alike.

Although the study has been carried out well, there are still some considerations that should be taken into account:

It is not clear from the results if the stimulation of macrophages with iron-depleted *Candida* has also been carried out with iron-depleted macrophages. If not, it is worth considering performing some of these experiments as a side-by-side comparison, as these cells could provide a source of iron with subsequent impacts on the fungal responses.

Related to this, how rapidly does the move to iron replete medium change the cell phenotype and vice versa? What level of iron is the threshold for this conversion?

The whole theory and premise of this study is that the fungal PAMPs are the only factor in these responses. Whilst these fungal PAMPs may be important, and even play the dominant role, other factors are also critical during in vivo, or even in vitro infections. Thus, it would be interesting to know what the impact of iron depletion is on other virulence/pathogenic factors, such as adhesions (e.g. ALS3, HWP1) and virulence factors (e.g. ECE1).

Does the *sef1* null mutant (and the *ftr1* and PKA null mutants) have problems/deficiencies in other areas, such as production of virulence and growth factors.

Finally, it would be of interest to know what further impact iron depletion has on other responses to the fungus – i.e. adhesion to and uptake by macrophages.

REVIEWER #1

1. They refer to unmasking, or masking but in their images the changes appear to be primarily at the poles which suggests that this is occurring at the bud scars. Is it the case that the changes in unmasking are primarily revolving around the bud scars? They should use calcofluor white staining to address this and determine if their changes are primarily seen associated with the bud scars.

Good question! It's true to say that bud scars are the main β -glucan exposing features on the *C. albicans* cell surface. However, we and others [e.g. *EMBO J* 24, 1277] also observe smaller punctate spots of β -glucan exposure that decorate the cell surface. The intensity of Fc-Dectin-1 staining decreases on both bud scars and punctate spots during masking. To illustrate this we have replaced Figure 4c with high resolution images of masked and unmasked *C. albicans* co-stained with Calcofluor White (CFW) and Fc-Dectin-1. We have added this information to the text (lines 164-166; 471-474).

2. In Figure 9A they conclude the pathways are acting in parallel in part because they cannot suppress the impact of *ftr1* $\Delta\Delta$ by adding di-cyclic-cAMP to the media. However, there is not a positive control for this experiment which makes it hard to know if the cAMP is having an effect.

This positive control was included with the original experiment, and these data are now included in the Supplementary Information (Fig. S2). The control confirms that, although this batch of db-cAMP did not suppress the iron masking defect of *ftr1* Δ cells (Fig. 9a), it was able to suppress masking defects in other mutants (Fig. S2). The data are discussed in the text (lines 257-260).

Please note that this control exploited the hypoxia-induced masking defect of *goa1* Δ cells [*mBio* 9, e01318-18]. This is because iron limitation-induced β -glucan masking is not dependent on adenylyl cyclase (Fig. 10).

3. To further support the parallel argument, they should test to see if they can suppress the masking impact of *tpk1* $\Delta\Delta$ *tpk2* $\Delta\Delta$ by overexpressing *Sef1*.

We attempted to test whether *SEF1* overexpression suppresses the masking defect of ability of *tpk1* Δ *tpk2* Δ cells. To achieve this we constructed Clp-based plasmids containing either ectopically expressed *SEF1* or GFP (as a control). However, our five attempts to transform this *tpk1* Δ *tpk2* Δ strain with either of these plasmids have all failed, whilst parallel transformations into control strains succeeded. Therefore, due to technical limitations that we have been unable to resolve, unfortunately we have been unable to test this.

4. In Figure 6 they measure the cell wall thickness. In Figure 9 they measure chitin production by calcofluor white staining and then imply that this indicates cell wall increases, but they need to measure this by TEM as they did in Figure 6 to be sure of this.

As requested, we have examined the thickness of the chitin-containing layer in the cell wall. We did not use TEM, as suggested, because this does not highlight the chitin distributed within the inner cell wall (e.g. Fig. 4). Instead, we quantified the intensity of Calcofluor White staining across masked and unmasked cells by fluorescence microscopy. This was done using the same cell populations that were analysed by flow cytometry (Fig. 9c). These analyses confirmed that Calcofluor White fluorescence

levels correlate with, and are a robust metric for, cell wall thickness in our experiments. These new data are included in Fig. 9b and discussed in lines 268-275. The old fluorescence micrographs have been moved to Fig. S3.

5. Iron depletion causes masking, but zinc depletion causes unmasking in their hands, yet both are part of nutritional immunity. They should address this paradox in their discussion.

Interesting point. Although nutritional immunity limits the availability of both iron and zinc, the adaptive responses to iron- and zinc-limitation are clearly different. This is not just reflected in their contrasting cell wall phenotypes, but in other (morphological) phenotypes. This is discussed on lines 333-339.

REVIEWER #2

Through phenotypic analyses of several deletion mutant strains, the authors convincingly demonstrate that the signaling pathway(s) involved in connecting iron availability to alterations in β -glucan exposure are not the same as the ones previously implicated in lactate- or hypoxia-mediated β -glucan modifications. However, in my opinion, it remains unclear how exactly iron depletion leads to β -glucan masking (i.e. the molecular mechanism). The authors do probe the involvement of well-known iron regulators but their analysis falls short of providing major mechanistic insights.

While we have not addressed masking mechanisms experimentally in this study, we do provide significant insight into the signalling mechanisms that underlie iron limitation-induced β -glucan masking. The molecular mechanisms that mediate β -glucan masking must involve mannan, but are clearly complex. We know from experience that these mechanisms are unlikely to be elaborated simply by analysing sets of mannan mutants, for example [*Nat. Microbiol.* 2, 16238]. These mechanisms will require detailed dissection and a separate study. Nevertheless, in light of the Reviewer's comment, we have now expanded our discussion of the molecular mechanisms on lines 374-387.

In terms of presentation of the story, I would recommend that the authors consider three points:

1. First, I think the authors overemphasize the "relevance" of the nutrients and other *in vitro* conditions that they evaluate to the environment inside the host. The fact of the matter is that we do not know for sure what signals or cues are really triggering what responses in *C. albicans* when it is within host tissues (multiple signals likely act simultaneously producing complex responses). The paper reads a lot like if it was a given that lactate or iron are responsible for *Candida*'s defense against immune cells. They may be part of the answer but it is far from definitive.

We are acutely aware that host niches are complex. Indeed, we continue to dissect the complexity of *C. albicans* responses to combinatorial environmental inputs *in vitro* [e.g. *mBio* 5, e01334-14; *mBio*. 7, e00331-16]. As stated, in this paper we focussed on signals that are known to trigger strong adaptive responses that promote fungal colonisation. Iron limitation is one of these [*Science* 288, 1062]. We suggest that, given the known significance of iron limitation *in vivo*, the masking phenotype is likely to have a significant influence on host-fungus interactions (lines 391-393). This is certainly the case for hypoxia-induced β -glucan masking [*mBio* 9, e02120-18.].

Nevertheless, to address the Reviewer's point, we have softened our closing conclusions (lines 393-397).

2. Second, I find the use the word 'anticipatory' (lines 32, 53, 64, 319) troubling. I am aware that similar ideas have been posited to explain the behavior of some bacterial pathogens, but still one needs much imagination to conceive that a unicellular organism is actually able to "anticipate" something and get ready for a particular event. It is simply not a precise expression.

We are not the first to use the term "anticipatory" in terms of regulatory responses [e.g. *Nature* 460, 220; *Mol. Biol. Evol.* 30, 573], and anticipatory responses in fungi were described ten years ago [*Nature* 460, 220; *Mol. Biol. Cell* 20, 4845]. Since we first described an anticipatory response in *C. albicans* a decade ago [*Mol. Biol. Cell* 20, 4845], diverse examples of anticipatory responses in fungal pathogens have started to emerge [*PLoS Pathog* 10, e1004356], and it has been shown that anticipatory responses can evolve rapidly in fungi [*Genome Biol. Evol.* 9, 1616]. We argue that these anticipatory responses influence host-fungus interactions [*Trends Microbiol* 27, 219]. No change.

3. Third, the changes in β -glucan exposure—which the authors emphasize—appear to be only one of multiple cell wall alterations induced by iron limitation. In fact, the data presented (Fig. 4) demonstrates that iron availability alters significantly the overall structure of the *C. albicans* cell surface. Hence, contrary to what the manuscript implies, it is not completely clear that the reduction in phagocytosis of yeast cells grown under iron limitation (Fig. 5) is due exclusively to β -glucan masking. What the authors present is a correlation, which is fine. But this caveat should be acknowledged more openly and widely in the manuscript.
Good point! We now highlight this caveat more clearly in the text (lines 309-312).
4. Minor points:
Lines 87-89: Format of references is incorrect.
Line 251: Typo: The authors probably mean 'previously' instead of 'preciously'
We apologise for these errors, which have now been corrected (lines 83 and 256).

REVIEWER #3

1. It is not clear from the results if the stimulation of macrophages with iron-depleted *Candida* has also been carried out with iron-depleted macrophages. If not, it is worth considering performing some of these experiments as a side-by-side comparison, as these cells could provide a source of iron with subsequent impacts on the fungal responses.
We apologise for the lack of clarity. Our experimental approach is now explained more clearly in the Results (lines 178-185) and Methods (lines 502-507). Our primary focus was the impact of fungal adaptation upon host recognition. Therefore, while it would be interesting to examine the effects of iron limitation upon macrophage functionality, this was not tested in our study.
Related to this, how rapidly does the move to iron replete medium change the cell phenotype and vice versa? What level of iron is the threshold for this conversion?
Interesting questions. We have now shown that β -glucan masking is triggered below an iron concentration threshold of about 10 μM Fe^{3+} (new Figure S1A), and also that

the reversion to β -glucan exposure following iron supplementation mirrors the growth of new *C. albicans* (new Figure S1B). These new data are discussed on lines 169-175.

2. The whole theory and premise of this study is that the fungal PAMPs are the only factor in these responses. Whilst these fungal PAMPs may be important, and even play the dominant role, other factors are also critical during *in vivo*, or even *in vitro* infections. Thus, it would be interesting to know what the impact of iron depletion is on other virulence/pathogenic factors, such as adhesions (e.g. ALS3, HWP1) and virulence factors (e.g. ECE1).

Good point. Transcript profiling has revealed that, as well as genes encoding iron assimilation functions and haem- and iron-containing proteins, additional loci are responsive to iron depletion in *C. albicans* [*Molec Microbiol* 53, 1451]. Genes involved in central metabolism, oxidative stress responses and the cell wall were highlighted. The cell wall genes were dealt with above, in terms of masking mechanisms (please see Reviewer #2, Opening Point). To address this Reviewer's point, we also highlight the multifactorial nature of host-fungus interactions, and now mention that iron limitation also affects some other virulence-related processes (lines 392-397).

Regarding the specific genes mentioned by the Reviewer, these are all hypha-specific (i.e. they are expressed in hyphae, not yeast cells). All of our experiments were performed on yeast cells to circumvent the significant practical difficulties inherent in analysing hyphae by cytometry. Furthermore, these genes were not highlighted by the transcript profiling study mentioned above [*Molec Microbiol* 53, 1451].

3. Does the *sef1* null mutant (and the *ftr1* and PKA null mutants) have problems/deficiencies in other areas, such as production of virulence and growth factors.

C. albicans sef1 and *sfu1* cells are defective in growth on iron depleted and replete media, respectively [*Cell. Host Microbe* 10, 118]. (Please note that we used iron limiting media, not iron depleted medium: see Methods.) Susan Noble's analysis of Sef1 regulated genes by transcript profiling and chromatin immunoprecipitation did not highlight virulence-related functions, and *HWP1*, *ECE1*, *ALS* or *SAP* genes were not present in the Sef1 regulon [*Cell. Host Microbe* 10, 118].

To our knowledge, transcript profiling has not been performed on *C. albicans ftr1* cells. PKA inactivation affects numerous process related to virulence in *C. albicans*, such as filamentation, adhesion, metabolism and stress responses [*Mol. Microbiol.* 105, 46; *Yeast* 26, 273]. These phenotypes are clearly relevant *in vivo*. However, our *in vitro* experimental design has allowed us to parse out the contribution of PKA to β -glucan masking (Fig. 6b). The same is true for Sef1 and Ftr1 (Fig. 8). We do not analyse these mutants *in vivo* or *ex vivo*. No change.

Finally, it would be of interest to know what further impact iron depletion has on other responses to the fungus – i.e. adhesion to and uptake by macrophages.

Iron depletion exerts minimal effects on the expression of adhesin genes [*Molec Microbiol* 53, 1451]. Adhesion itself was not tested. However, our data on the impact of fungal iron adaptation on macrophage uptake and cytokine responses were presented in Fig. 5. Effects of iron limitation on the expression of virulence-related genes are now mentioned on lines 394-396.

REVIEWERS' COMMENTS:

Reviewer #2 (Remarks to the Author):

[No further comments for author.]

Reviewer #3 (Remarks to the Author):

All my questions have now been answered well, and the manuscript is significantly improved. All reviewers' points and queries have been completely addressed. As such, this article is now ready for publication.